# Comparative Analysis of the Characteristics, Phylogenetic Relationships of the Complete Chloroplast Genome, and Maternal Origin Track of White Poplar Interspecific Hybrid GM107

Bin Guo [1,2,†], Tingting Chen [1,†], Ying Li [1], Shanwen Li [3], Wasif Ullah Khan [1], Ren-Gang Zhang [4], Kai-Hua Jia [5] and Xinmin An [1,*]

[1] State Key Laboratory of Tree Genetics and Breeding, National Engineering Research Center of Tree Breeding and Ecological Remediation, Beijing Advanced Innovation Center for Tree Breeding by Molecular Design, College of Biological Sciences and Technology, Beijing Forestry University, Beijing 100083, China
[2] Shanxi Academy of Forestry and Grassland Sciences, Taiyuan 030012, China
[3] Shandong Academy of Forestry, Jinan 250014, China
[4] Yunnan Key Laboratory for Integrative Conservation of Plant Species with Extremely Small Populations/ Key Laboratory for Plant Diversity and Biogeography of East Asia, Kunming Institute of Botany, Chinese Academy of Sciences, Kunming 650201, China
[5] Key Laboratory of Crop Genetic Improvement & Ecology and Physiology, Institute of Crop Germplasm Resources, Shandong Academy of Agricultural Sciences, Jinan 250100, China
* Correspondence: anxinmin@bjfu.edu.cn
† These authors contributed equally to this work.

**Abstract:** White poplars are a dominant tree species in natural ecosystems throughout China, where they are also widely planted for agroforestry and industrial uses. However, the phylogenetic relationships among poplars in section *Populus* have a complex genetic background influenced by frequent hybridization events and, to date, only a few studies have attempted to clarify this background. In this study, we performed de novo assembly of the whole chloroplast (cp) genome of an elite individual GM107 with the size of 156,493 bp, which consists of a large single-copy (LSR) (84,610 bp), a small single-copy (SSC) (16,567 bp), and a pair of inverted repeats (27,658 bp). It comprises 127 genes, including 85 protein-coding genes, 36 tRNAs, and 6 rRNAs. Comparative analysis of the cp genomes was conducted among 7 poplars in section *Populus* and 4 cp DNA markers with >1% variable sites were detected. We found that *Populus alba* was the most closely related species to GM107 by phylogenetic analyses. RNA sequencing detected 66 genes that participated in translation, transcription, and photosynthesis. The expression levels of almost all 66 genes were higher in leaves than in other tissues, except for *PtatpF* and *PtatpH*. In all tissues, we detected higher transcript abundances of *PtndhF*, *PtpsbA*, *PtpsbB*, *Ptrps14*, *PtatpF*, and *PtatpH* than of other genes. Both cp genome and transcriptome data help understand evolution events in section *Populus* and unravel the origin of Chinese white poplars, and may contribute to the molecular genetic improvement of wood properties and carbon sink capacities in the breeding of poplars in this section.

**Keywords:** chloroplast genome; phylogenetic analysis; white poplars; transcriptome





## 1. Introduction

Genus *Populus*, which consists of nearly 30 species with abundant natural distribution in the Northern Hemisphere, is taxonomically classified into six sections: *Populus*, *Tacamahaca*, *Aigeiros*, *Turanga*, *Leucoides*, and *Abaso*. The *Populus* section, comprised of white poplar, includes *Populus tremuloides*, *Populus davidiana*, *Populus tremula*, *Populus adenopoda*, *Populus sieboldii*, *Populus alba*, and *Populus grandidentata.* These species are often segregated into geographical and morphological subspecies and varieties due to their wide natural

distribution [1]. Their capacity for outcrossing and wind pollination, as well as the absence of barriers to crossbreeding, has led to numerous hybrids [2].

Chinese white poplar is indigenous to China, with a wide distribution range [3]. It exhibits excellent features, such as rapid growth, a thick and straight trunk, strong resistance to environmental stresses, and a long lifespan (typically 100–200 years, but reaching 500 years). It is an industrial tree species, providing timber, pulp, and biofuel [4]. Thus, it is more economically, ecologically, and evolutionarily valuable than the other white poplars found in China [5].

Interspecific and intraspecific hybridization occurs frequently among white poplars due to long-distance pollination by wind and overlapping distribution ranges over large geographical extents [6]. Therefore, Chinese white poplar has a complex genetic background. It was originally thought to be a distinct species in the *Populus* section [2], but is now considered to be a hybrid of *P. alba* and *P. adenopoda* or a tri-hybrid of both species with *P. davidiana* [7,8]. Recent evidence indicated that another white poplar elite individual GM15 originated from a hybridization event of *P. adenopoda* and *P. alba* var. *pyramidalis* approximately 3.93 Mya [5], which suggests that the origin of Chinese white poplar is more complicated than previously described. The development of third-generation sequencing technology has significantly improved the study of poplar phylogeny. Analyzing the cp genome through phylogenetic analysis is a convincing method to unravel the mystery of hybrid origins.

Regarding the photosynthetic organelle, the chloroplast (cp) genome consists of circular DNA molecules that are highly similar and range in size from 115 to 165 kb [6,9]. The cp has a conserved genomic structure, gene sequence, and gene type, and consists of a large single-copy (LSC) region and a small single-copy (SSC) region separated by a pair of inverted repeats (IRa and IRb) [10,11]. The cp genome is a metabolic center with highly conserved physiological cell functions, especially for genes related to photosynthesis [9,12]. The cp genome is characterized by a slow molecular evolution rate and follows the single parental inheritance principle. Therefore, cp genomes are widely used in population taxonomy and phylogeny, genetics, and evolution studies [13,14]. The parental origin of hybrids with complex genetic backgrounds can be more accurately traced using cp genomes, which are considered more valuable and reliable markers than any others. With the advancement of third-generation sequencing technology, cp genomes are becoming increasingly popular.

In our study, we conducted a comprehensive analysis by performing de novo assembly of the cp genome of an elite individual GM107 and comparative analysis with seven other white poplar species to reveal its unique structures, variations, and repeat sequences, and to elucidate the phylogenetic relationships among white poplar species. In addition, we performed transcriptome sequencing and examined the gene expression profiles related to photosynthesis, transcription, and translation of cps in the roots, stems, leaves, and petioles of four poplar species. This study will offer substantial proof of the ancestry of the Chinese white poplar and aid in the preservation and optimization of its genetic resources.

## 2. Materials and Methods

### 2.1. Plant DNA Extraction and Sequencing

We collected young leaf tissue samples from 1-month-old tissue-cultured plantlets of an elite individual GM107 (male clone) and extracted genomic DNA immediately after sampling using the Qiagen DNeasy Plant Mini Kit (Qiagen, Hilden, Germany). The quality of the genomic DNA was visualized by running agarose gel and quantified using a NanoDrop spectrophotometer (Thermo Fisher Scientific, Waltham, MA, USA) to determine its integrity, purity, and concentration. The short-insert polymerase chain reaction (PCR) free genomic DNA library (300–500 bp) was constructed according to the manufacturer's instructions (Illumina, San Diego, CA, USA) for paired-end sequencing. A 20 kb genomic DNA library was constructed for single-molecule, real-time (SMRT) sequencing according to the manufacturer's instructions (Oxford Nanopore Technologies (ONT), Oxford,

UK). Finally, sequencing was conducted on the Illumina HiSeq X Ten sequencer on the ONT platform.

### 2.2. Plant RNA Extraction and Transcriptome Sequencing

Tissue samples from the roots, stems, leaves, and petioles were collected from 4 1-month-old tissue-cultured plantlets of *P. tomentosa* GM15, an elite individual GM107, *P. alba* var. *pramidalis*, and *P. trichocarpa*, and treated instantly with liquid nitrogen for RNA extraction. Four replicates were obtained for each sample. Total RNA was extracted with TRIzol reagent and mRNAs were refined through NEBNext Ultra RNA Library Prep Kit (New England Biolabs, Ipswich, MA, USA). Two micrograms of RNA from each sample were used to prepare the RNA sequencing libraries and then sequenced on the DNBSEQ platform. Salmon v1.6.0 software [15] was used to acquire transcript abundances (transcripts per million; TPM) for each sample and the number of reads originating from the transcripts was estimated. Additionally, Tximport software package was utilized to prepare inputs for expression profile analysis [16]. We used the mean of the four replicates to scan for expressed genes (TPM > 1 in at least 1 tissue), and then normalized the data ($\log_{10}$(TPM + 1)) to draw heatmaps using the *pheatmap* package in R software (R Core Team, Vienna, Austria).

### 2.3. Chloroplast Genome Assembly and Annotation

After the filtering process, clean reads were produced, including those from nuclear and cp genomes. Reads from the cp genome were obtained by alignment with 14 related species cp genomes (including *Populus trichocarpa* NC_009143.1, *Populus balsamifera* NC_024735.1, *P. alba* NC_008235.1, *P. tremula* NC_027425.1, *P. tremula* × *P. alba* NC_028504.1, *P. davidiana* NC_032717.1, *Populus lasiocarpa* NC_036040.1, *P. adenopoda* NC_032368.1, *Populus rotundifolia* NC_033876.1, *Populus fremontii* NC_024734.1, *Populus qiongdaoensis* NC_031398.1, *Populus ilicifolia* NC_031371.1, and *Populus euphratica* NC_024747.1) using Graphmap software [17]. The CANU de novo assembler was utilized to assemble the obtained reads [18]. Subsequently, the primary assembly was polished by ONT sequencing using next-generation sequencing data [19]. The cp genome was thoroughly annotated using the GeSeq software, utilizing its default parameters as specified on the Chlorobox website (https://chlorobox. mpimp-golm.mpg.de/geseq.html, accessed on 18 December 2022). The visualization of circular chloroplast genome maps was accomplished through the utilization of the Organellar Genome DRAW software [20]. The cp sequence generated in this study was submitted to the National Center for Biotechnology Information (NCBI) database with Genebank accession no. MK251149.

### 2.4. Comparative Genomic Analyses

We downloaded cp genome sequences of *Salix suchowensis* and 7 *Populus* species (*P. davidiana*, *P. tremula*, *P. alba*, *P. adenopoda*, *P. rotundifolia*, *P. tremula* × *P. alba*, and *P. alba* × *Populus glandulosa*) published in genebank. Using the mVISTA program (http://genome. lbl.gov/vista/mvista/submit.shtml, accessed on 18 December 2022), we compared the complete plastid genomes of these *Populus* species to the annotated *P. tomentosa* cp genome as a reference [6], and investigated the regions of high divergence as well as variable and parsimony informative sites across the complete cp genomes. Additionally, the LSC, SSC, and IR regions of 7 taxa were identified using DnaSP v5.0 software [21]. The codon usage of the elite individual GM107 cp genome was analyzed using the CodonW program in MEGA 7.0 [22].

### 2.5. Characterization of Repeat Sequences and SSRs

The numbers of large repeats, including forward, reverse, palindromic, and complementary repeats, were identified using the REPuter program [23] based on the following criteria: sequence identities of 90%, cutoff point of ≥30 bp, Hamming distance of 3, and minimum repeat size of 30 bp. Furthermore, the Tandem Repeats Finder online program was

utilized to identify tandem repeat sequences with a size of more than 10 bp, using default parameters [24] Lastly, SSRs were identified through MISA Perl script [25], with a minimum threshold of 10 repeat units for mono-, di-, tri-, tetra-, penta-, and hexanucleotides.

*2.6. Phylogenetic Analyses*

The phylogenetic analysis was performed using the complete cp genomes of 7 Populus species (*P. davidiana*, *P. tremula*, *P. alba*, *P. adenopoda*, *P. rotundifolia*, *P. tremula* × *P. alba*, and *P. alba* × *P. glandulosa*), where *Salix suchowensis* (NC_026462.1) was used as the outgroup. Complete cp genome alignment was performed using MAFFT v.7 software [26] and manually edited as required. Subsequently, phylogenetic analysis was conducted using maximum likelihood (ML) methods based on the 8 cp whole-genome sequence alignments. In order to generate a robust phylogenetic tree, the ML analyses were conducted using IQ-tree Inference of Quantitative Phylogenies Tree software [27] with 10,000 ultrafast bootstrap replicates [28].

## 3. Results

*3.1. Cp Genome Structure*

We conducted de novo assembly of the complete cp genome of the elite individual GM107. The primary assembly generated from ONT data was polished using Illumina data, resulting in a final cp genome assembly of 156,493 bp (Figure 1, Table 1).

**Table 1.** Summary of complete chloroplast genome of the elite individual GM107.

| Characteristics | Elite Individual GM107 |
| --- | --- |
| Length (bp) | 156,493 |
| GC content (%) | 36.74 |
| AT content (%) | 63.26 |
| LSC GC content (%) | 39.31 |
| SSC GC content (%) | 30.49 |
| IR GC content (%) | 41.95 |
| LSC length (bp) | 84,610 |
| SSC length (bp) | 16,567 |
| IR length (bp) | 27,658 |
| CDS length (bp) | 80,562 |
| tRNA length (bp) | 2648 |
| rRNA length (bp) | 8599 |
| Gene number | 127 |
| Gene number in IR regions | 34 |
| Protein-coding gene number | 85 |
| Protein-coding gene (%) | 66.93 |
| rRNA gene number | 6 |
| rRNA (%) | 4.72 |
| tRNA gene number | 36 |
| tRNA (%) | 28.35 |

The chloroplast genome features a circular molecular structure composed of three regions. The largest region, referred to as the large single-copy (LSC) region, spans 84,610 base pairs and makes up 54.1% of the genome. The second region, the small single-copy (SSC) region, is much smaller, making up 10.6% of the genome, consisting of 16,567 base pairs. The remaining 17.7% of the genome is divided into two intergenic regions (IRa/b), each with a size of 27,658 base pairs (Table 1).

The cp genome contains 127 genes, including 85 protein-coding genes (66.93% of the total), 36 tRNA genes (28.35%), and 6 rRNA genes (4.72%). The noncoding structures, such as intergenic spacers and introns, occupy the remaining portion. The entire genome has GC content of 36.5% with increased GC concentration (41.95%) in the IR regions, followed by LSC (39.31%) and SSC (30.49%), respectively (Table 1).

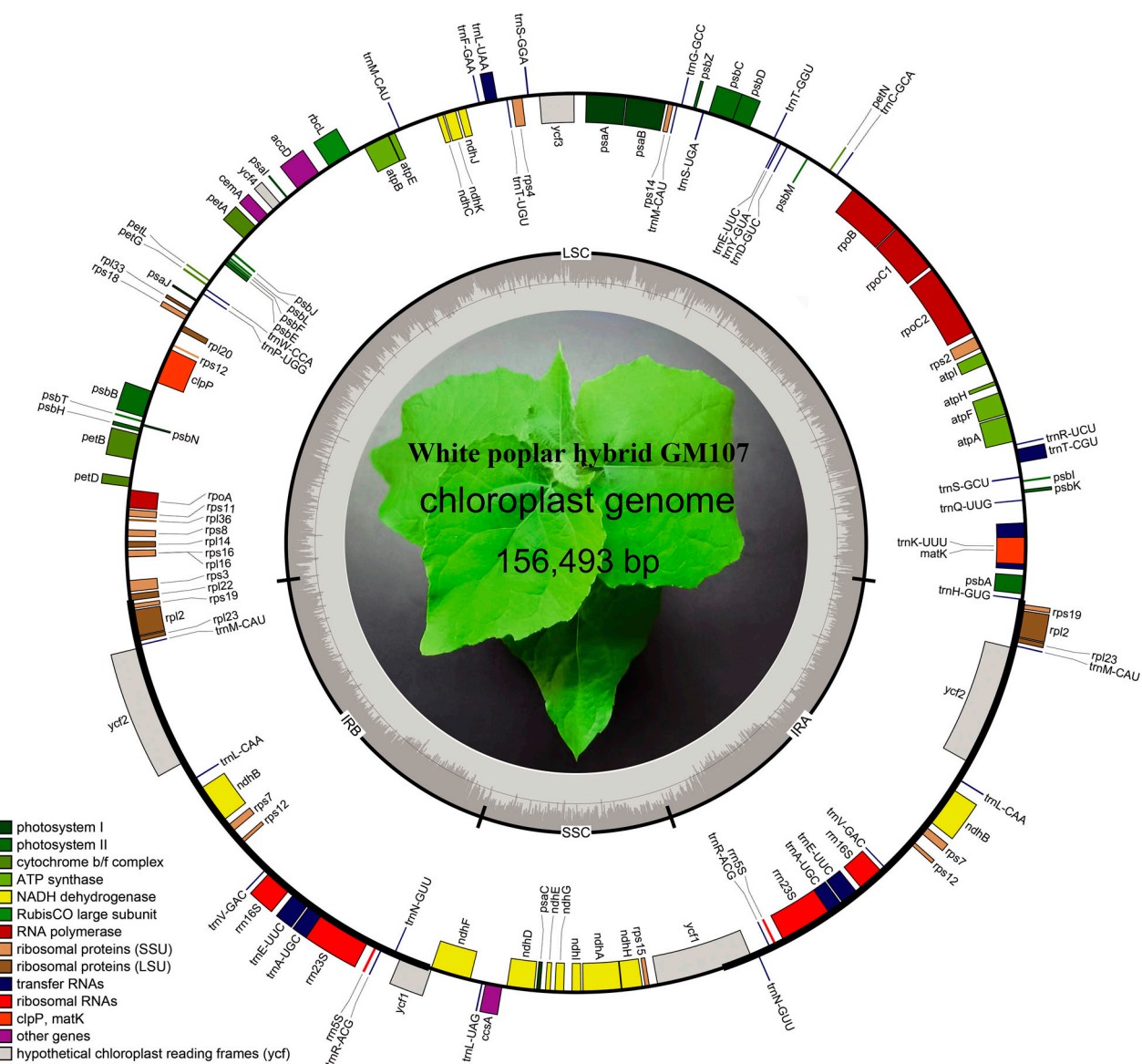

**Figure 1. Gene circle map of the elite individual GM107 chloroplast (cp) genome.** The colored bars in the diagram represent various functional groups. The bold lines of the main circle outline the extent of the inverted repeat regions (IRA and IRB), dividing the genome into small single-copy (SSC) and large single-copy (LSC) regions. Genes located on the inner and outer parts of the main circle are transcribed in a clockwise and counterclockwise direction, respectively. The inner circle features dark gray and light gray columns that denote the GC and AT content of the DNA, respectively.

All genes were classified into four categories: photosynthesis-performing genes, transcription-associated and translation-associated genes, genes related to RNA and its processing, and other genes. A total of 17 genes possessed introns, with 15 of these genes (comprising 9 protein-coding genes and 6 transfer RNA genes) possessing a single intron each, while *clpP* and *ycf3* possessed 2 introns each. Additionally, 18 genes were duplicated in the IR regions (Table 2). These findings align with those reported in other *Populus* chloroplast genomes [5,29,30].

**Table 2.** List of annotated genes in the chloroplast genome of the elite individual GM107.

| Category of Genes | Family Name List of Genes | List of Genes |
|---|---|---|
| Photosynthesis-related genes | Subunit of rubisco<br>Subunits of photosystem I<br>Assembly of photosystem I | *rbcL*<br>*psaA, psaB, psaC, psaI, psaJ*<br>*ycf4, ycf3* ** |
| | Subunits of photosystem II | *psbA, psbB, psbC, psbD, psbE, psbF, psbI, psbJ, psbK, psbL, psbM, psbN, psbT, psbZ* |
| | Subunits of ATP synthase | *atpA, atpB, atpE, atpF* *, *atpH, atpI* |
| | Subunits of cytochrome b/f complex | *petA, petB* *, *petD, petG, petL, petN* |
| | c-type cytochrom synthesis gene | *ccsA* |
| | Subunits of NADH-dehydrogenase | *ndhA* *, *ndhB* * (×2), *ndhC, ndhD, ndhE, ndhF, ndhG, ndhH, ndhI, ndhJ, ndhK* |
| | Subunits of protochlorophyllide reductase | - |
| Transcription-related and translation-related genes | DNA-dependent RNA polymerase | *rpoA, rpoB, rpoC1, rpoC2* |
| | Small subunit of ribosome | *rps11, rps12* * (×2), *rps14, rps15, rps16, rps18, rps19* (×2), *rps2, rps3, rps4, rps7* (×2), *rps8* |
| | Large subunit of ribosome | *rpl14, rpl16, rpl2* * (×2), *rpl20, rpl22, rpl23*(×2), *rpl33, rpl36* |
| | Translational initiation factor | - |
| | Elongation factor | - |
| RNA and its processing | rRNA genes | *rrn16S* (×2), *rrn23S* (×2), *rrn5S* (×2) |
| | tRNA genes | *trnH-GUG, trnK-UUU* *, *trnQ-UUG, trnS-GCU, trnT-CGU* *, *trnR-UCU, trnC-GCA, trnD-GUC, trnY-GUA, trnE-UUC* *, *trnT-GGU, trnS-UGA, trnG-GCC, trnM-CAU, trnS-GGA, trnT-UGU, trnL-UAA* *, *trnF-GAA, trnM-CAU, trnW-CCA, trnP-UGG, trnM-CAU* (×2), *trnL-CAA* (×2), *trnV-GAC* (×2), *trnE-UUC* (×2), *trnA-UGC* * (×2), *trnR-ACG* (×2), *trnN-GUU* (×2), *trnL-UAG* |
| Other genes | RNA processing | *matK* |
| | Subunit of Acetyl-CoA-carboxylase | *accD* |
| | Envelop membrane protein | *cemA* |
| | Protease | *clpP* ** |
| | Component of TIC complex | *ycf1* (×2) |
| | Hypothetical proteins | *ycf2* (×2) |

* Genes with one intron; ** genes with two introns; (×2) genes with duplication in the IR regions.

### 3.2. Expansion and Contraction of IR Regions

Comparative sequence analysis of eight white poplar species indicated a high degree of conservation in the chloroplast genome structure, gene number, and sequence. Nevertheless, variations in structure and size at the IR boundaries were noted (as shown in Figure 2). The boundary sites between regions are designated as JLB (IRb/LSC), JSB (IRb/SSC), JSA (SSC/IRa), and JLA (IRa/LSC). The genes located in the boundary regions consist of *rpl22*, *rps19*, *trnL*, *ycf1*, *trnN*, *ndhF*, *trnH*, and *ycf15*. All eight analyzed species had functional *ycf1* and *trnN* genes in the JSA region, and six species had *ycf1* and *ndhF* genes in the JSB region, while the other two had *trnN* and *ycf1* (GM107) and *trnN* and *ndhF* genes (*P. rotundifolia*).

Seven species had functional *rpl22* and *rps19* genes in the JLB region, and *rps19* and *trnH* in the JLA region. Only *P. rotundifolia* had *trnL* and *ycf15* in both JLB and JLA regions. The genes' placement in the JLB and JLA boundary regions was comparatively stable, while the JSB and JSA boundary regions exhibited significant variation across the eight species. Such differences in the chloroplast (cp) genome often result in expansion and contraction, leading to divergence events between species [9].

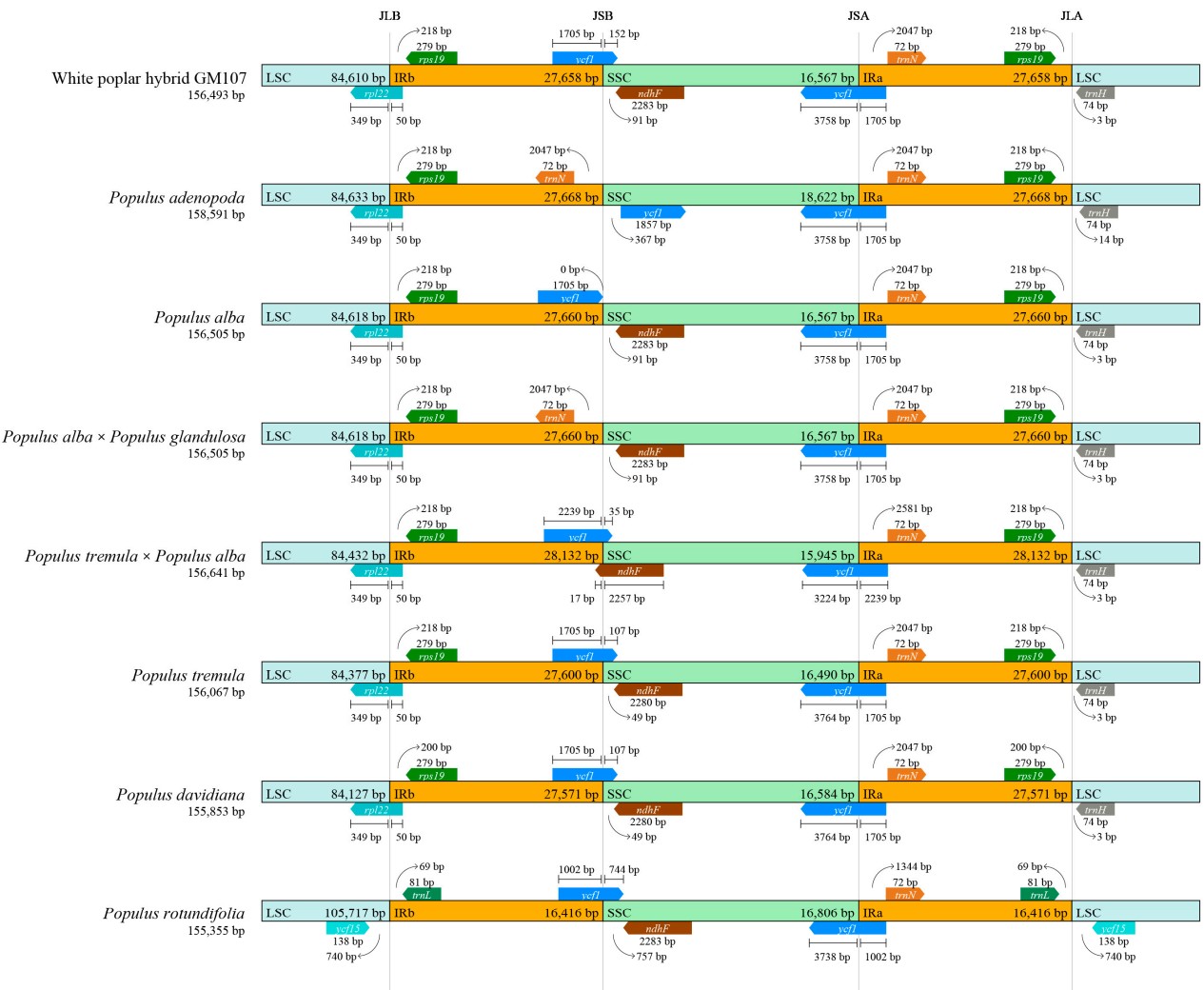

**Figure 2.** Comparison of LSC, IRs, and SSC region junction among the seven *Populus* cp genomes.

A common 50 bp extension to the IRb was detected in LSC gene *rpl22* of all species except *P. rotundifolia*. The SSC gene *ycf1* had a 1705 bp extension to the IRa regions of the elite individual GM107, *P. adenopoda*, *P. alba*, *P. alba* × *P. glandulosa*, *P. tremula*, and *P. davidiana*. In *P. rotundifolia* and *P. tremula* × *P. alba*, the SSC gene *ycf1* extended 2239 and 1002 bp to the IRa and IRb regions, respectively. The *ycf1* genes in the JSB border region presented some variation, with 35–744 bp extensions to the SSC region in *P. tremula* × *P. alba*, elite individual GM107, *P. tremula*, *P. davidiana*, and *P. rotundifolia*.

### 3.3. Repeat Structure and SSR Analyses

A comprehensive analysis of 8 chloroplast genomes revealed a total of 374 repeat sequences, with lengths varying from 30 to 76 base pairs. The majority of the repeats were found to be in the range of 30 to 42 base pairs (Table 3), and all repeat types (forward, reverse, palindromic, and complement) were found in each genome (Table 4). Repeats of

each type presented similar features across all eight cp genomes, except for complement repeats. *Populus alba*, *P. alba* × *P. glandulosa*, *P. tremula* × *P. alba*, and the elite individual GM107 had four complement repeats, while *P. rotundifolia*, *P. davidiana*, *P. adenopoda*, and *P. tremula* had two, one, zero, and zero complement repeats, respectively (Table 4). The forward repeats presented the highest proportion, accounting for 47.06% of all repeat types, followed by palindromic (36.10%), reverse (11.76%), and complement repeats (5.08%).

**Table 3.** Repeat length frequency among eight *Populus* chloroplast genomes.

| Species | Number of Repeats | | | | | | | | | | | | | | | | | | | | |
|---|---|---|---|---|---|---|---|---|---|---|---|---|---|---|---|---|---|---|---|---|---|
| | 30 | 31 | 32 | 33 | 34 | 35 | 36 | 37 | 38 | 39 | 40 | 41 | 42 | 44 | 45 | 46 | 47 | 48 | 55 | 68 | 76 |
| *P. adenopoda* | 13 | 4 | 4 | 1 | 3 | 1 | 1 | 0 | 0 | 6 | 0 | 1 | 6 | 0 | 1 | 0 | 0 | 1 | 1 | 1 | 1 |
| *P. alba* | 16 | 4 | 6 | 2 | 3 | 1 | 2 | 1 | 0 | 6 | 0 | 1 | 2 | 0 | 0 | 1 | 0 | 0 | 1 | 1 | 1 |
| *P. alba* × *P. glandulosa* | 16 | 4 | 6 | 2 | 3 | 1 | 2 | 1 | 0 | 6 | 0 | 1 | 2 | 0 | 0 | 1 | 0 | 0 | 1 | 1 | 1 |
| *P. davidiana* | 13 | 5 | 4 | 2 | 0 | 2 | 1 | 0 | 0 | 6 | 0 | 1 | 2 | 1 | 0 | 0 | 0 | 0 | 1 | 1 | 1 |
| *P. rotundifolia* | 12 | 8 | 11 | 3 | 0 | 2 | 1 | 1 | 1 | 6 | 2 | 1 | 4 | 1 | 0 | 0 | 1 | 0 | 1 | 1 | 1 |
| Elite individual GM107 | 16 | 4 | 6 | 2 | 3 | 1 | 2 | 1 | 0 | 6 | 0 | 1 | 2 | 0 | 0 | 1 | 0 | 0 | 1 | 1 | 1 |
| *P. tremula* | 13 | 3 | 3 | 1 | 1 | 2 | 1 | 1 | 0 | 7 | 0 | 2 | 2 | 0 | 0 | 1 | 0 | 0 | 1 | 1 | 1 |
| *P. tremula* × *P. alba* | 16 | 4 | 6 | 2 | 3 | 1 | 2 | 1 | 0 | 6 | 0 | 1 | 2 | 0 | 0 | 1 | 0 | 0 | 1 | 1 | 1 |

**Table 4.** Repeat type frequency among eight *Populus* chloroplast genomes.

| Species | Number of Repeats | | | |
|---|---|---|---|---|
| | Forward | Palindromic | Reverse | Complement |
| *P. adenopoda* | 23 | 19 | 3 | 0 |
| *P. alba* | 22 | 16 | 6 | 4 |
| *P. alba* × *P. glandulosa* | 22 | 16 | 6 | 4 |
| *P. davidiana* | 21 | 15 | 3 | 1 |
| *P. rotundifolia* | 25 | 20 | 10 | 2 |
| Elite individual GM107 | 22 | 16 | 6 | 4 |
| *P. tremula* | 19 | 17 | 4 | 0 |
| *P. tremula* × *P. alba* | 22 | 16 | 6 | 4 |
| Total | 176 | 135 | 44 | 19 |

A total of 1034 simple repeat sites were detected among the 8 *Populus* cp genomes, ranging from 126 to 135 SSRs per species. These repeats are comprised of a diverse set of one to six base pair units. Among these poplar species, the most abundant types of repeats were mononucleotides (101–106), followed by dinucleotides (8–10), trinucleotides (5–8), and tetranucleotides (8–1). Interestingly, only two species, *P. alba* and *P. rotundifolia*, had penta-base repeats, each with a single instance. There were one to two six-base repeats, except in *P. rotundifolia* (Figure 3a). In total, there were 1034 SSRs, among which 76.02% of loci were in intergenic spacer (IGS) regions, and 11.99% of which were distributed in introns and coding sequences (CDSs) (Figure 3b). In 8 species of *Populus*, the most abundant type of repeat unit found was mononucleotide A/T, with concentrations ranging from a high of 81.25% in *P. davidiana* to a low of 78.91% in *P. alba* × *P. glandulosa*. The presence of mononucleotide G repeats was limited to only *P. adenopoda*. Of the two-base pair repeats, AT was more commonly observed than TA. Additionally, the five-base repeat sequence ATAAT was found in *P. alba*, *P. alba* × *P. glandulosa*, and the elite individual GM107. Another five-base repeat identified was TAGAT, which was exclusive to *P. rotundifolia*. There were four six-base repeats: ATAAAA, ATATAG, ATCTAT, and TTTCTA. Among these, ATATAG and ATCTAT appeared in *P. adenopoda*, *P. alba*, *P. alba* × *P. glandulosa*, and the elite individual GM107; ATAAAA appeared in *P. davidiana*, *P. tremula*, and *P. tremula* × *P. alba*; and TTTCTA appeared only in *P. tremula* × *P. alba* (Figure 3a).

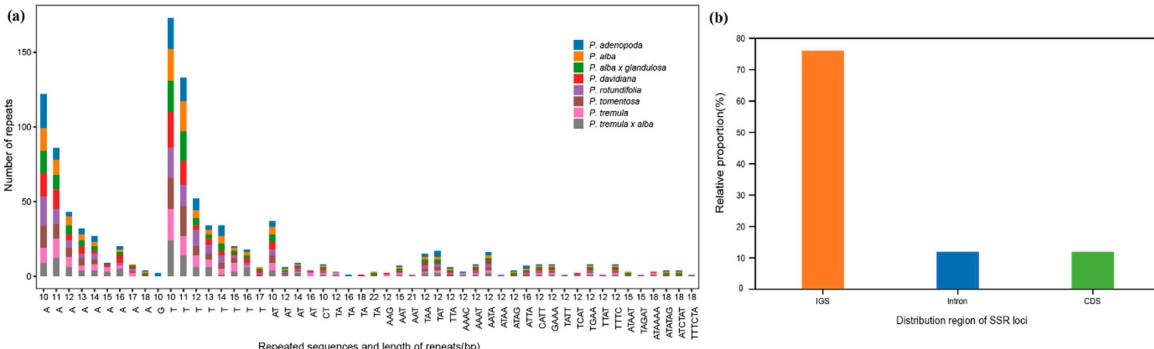

**Figure 3. The microsatellites and oligonucleotide repeat in the eight *Populus* cp genomes.**
(**a**) Numbers of repeats by length; (**b**) distribution of microsatellites.

### 3.4. Sequence Divergence Analyses

To assess the diversity among the chloroplast genome sequences of eight white poplar individuals, we used the mVISTA program to visualize the degree of identity between the sequences and highlight highly conserved regions with color (Figure 4). As depicted in Figure 4, the sequences exhibited high sequence similarity (>90%) and gene order conservancy among poplar species. Moreover, the coding regions were more conserved than the noncoding regions, which is consistent with previous findings [6,31,32]. However, the coding regions of the *ycf1* gene were the most divergent among *Populus* section derivatives, indicating its potential use as a barcode for taxonomic and phylogenetic analysis of *Populus* species. Some highly divergent noncoding regions included: *trnS-trnT*, *petN-psbM*, *trnT-psbD*, *ndhC-trnM*, and *psbE-trnP*.

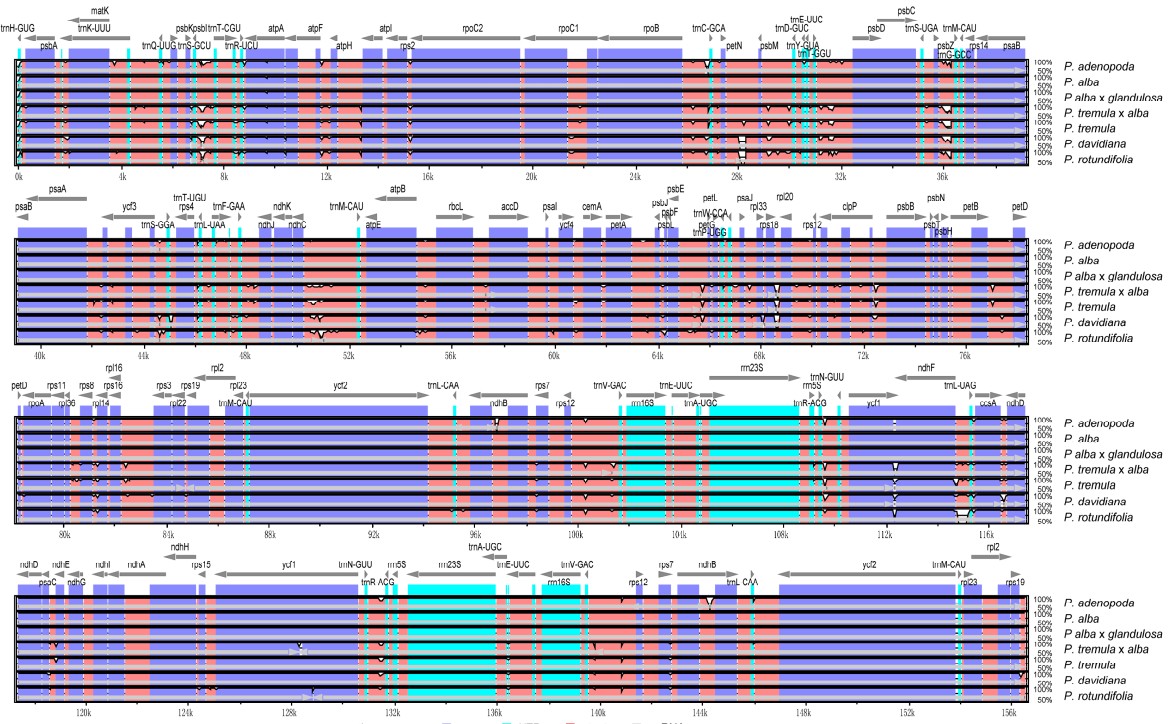

**Figure 4. Seven *Populus* cp genomes' comparison through mVISTA program.** White peaks indicate differences in genomics. The *y*-axis indicates the percentage identity (shown: 50–100%). Gray arrows and thick black lines above the alignment indicate gene orientation. Purple bars represent exons, blue bars represent untranslated regions (UTRs), pink bars represent noncoding sequences (CNSs), gray bars represent mRNA.

To gain a comprehensive understanding of sequence deviation, we conducted a sliding window analysis to examine the variation within nucleotides of eight chloroplast genomes. Figure 5 indicates that none of the CDS genes had Pi values above 0.02, and the intergenic sequences exhibited greater deviation than the gene regions. The figure clearly demonstrates that the SSC and LSC regions were more diverse compared to the IR regions. Intergenic sections with a Pi value greater than 1% included psbI-trnT-CGU (1.11%), ndhC (1.69%), petA-psbL (1.11%), and ycf1-ndhF (1.58%) (Figure 5).

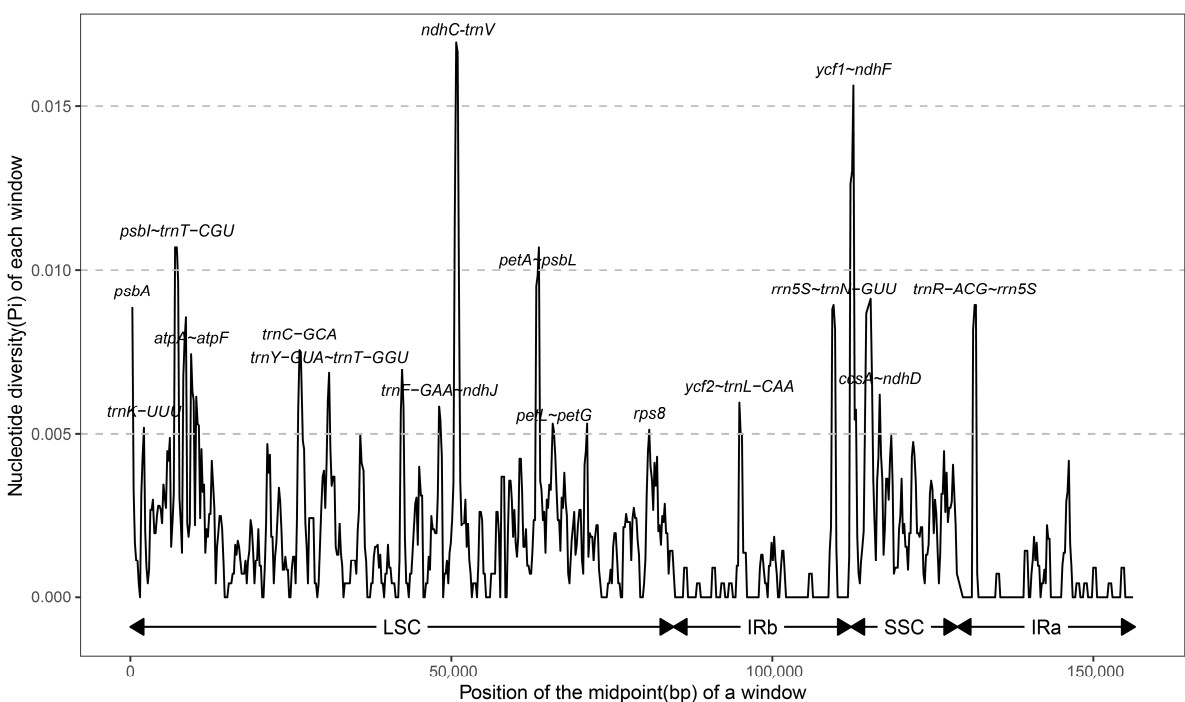

**Figure 5.** The nucleotide variability (Pi) values of eight *Populus* species.

### 3.5. Codon Biased Usage Analysis

The entire chloroplast genome of the elite individual GM107 contained 85 protein-coding genes, whose codons fit into 64 codon types. The analysis of amino acid rates exhibited that the top three most abundant amino acids were Leucine (Leu), Arginine (Arg), and Serine (Ser), with an occurrence of 9.4%. These were followed by Alanine (Ala), Glycine (Gly), Proline (Pro), Threonine (Thr), and Valine (Val), which had an occurrence of 6.3%. On the other hand, Tyrosine (Trp) and Methionine (Met) were the least abundant, with an occurrence of only 1.6%.

The calculation of the relative synonymous codon usage frequency was performed on protein-coding sequences derived from the cp genome of the elite plant individual GM107. The relative synonymous codon usage analysis, presented in Figure 6, revealed that 21 codons displayed RSCU values greater than 1, indicating a preference for certain codons. On the other hand, codons encoding Met and Trp had an RSCU value of one, indicating a lack of bias in usage. The Arg codon (AGA) had the highest RSCU value (2.02) among all codons, demonstrating a stronger preference compared to the other 5 Arg codons (AGG, CGA, CGC, CGG, and CGT).

Codon preference is prevalent in both prokaryotic and eukaryotic systems and understanding intraspecific or interspecific gene codon preference during species evolution is of significant importance [33]. The use of codons in gene translation is influenced by long-term natural selection, artificial domestication, and mutations. Highly expressed genes are mainly impacted by natural selection in determining their codon preference. Conversely, the preference of codons in genes with low expression is primarily determined by mutations [34].

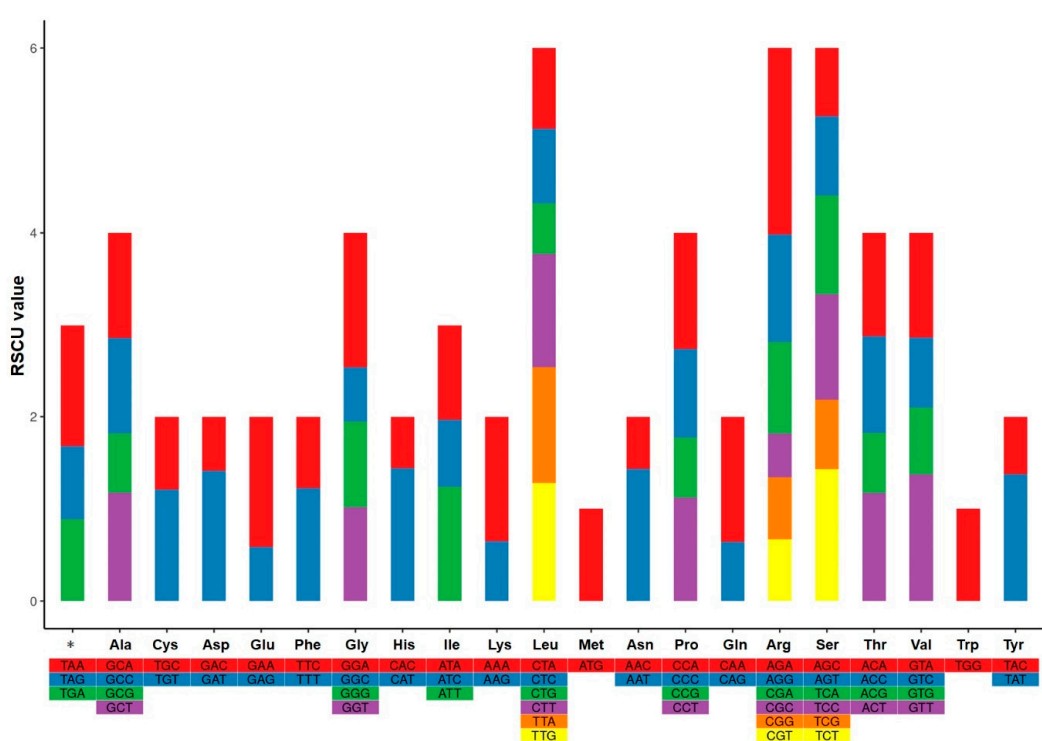

**Figure 6.** Relative synonymous codon usage and biased codon usage. * Stop code.

### 3.6. Phylogenetic Analyses

To determine the evolutionary relationship of the exceptional individual GM107 within the white poplar species, we chose seven additional cp genomes and utilized *Salix suchowensis* as a reference point. The maximum likelihood (ML) method was used to construct a phylogenetic tree, which is presented in Figure 7.

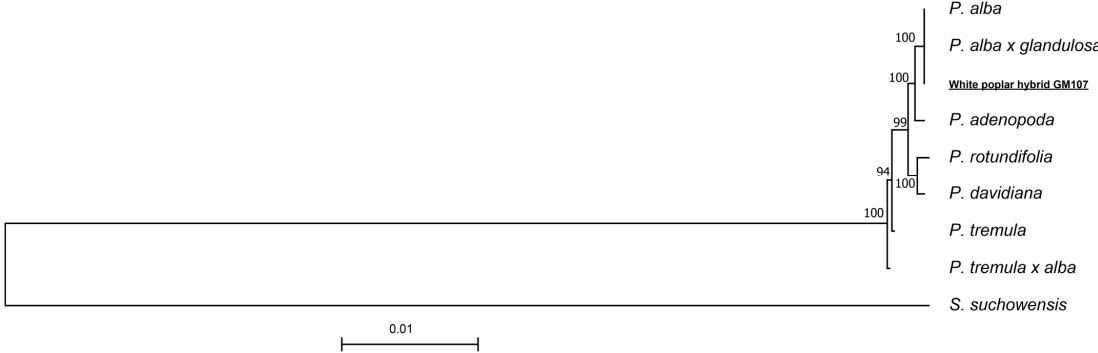

**Figure 7.** The phylogenetic tree of cp genomes from nine species. The tree was constructed using maximum likelihood. *S. suchowensis* was set as outgroup.

The study indicated that the divergence between *P. tremula* and other *Populus* species happened first, leading to the classification of other white poplars into two evolutionary branches. The first branch includes *P. rotundifolia* and *P. davidiana*, while the second branch contains *P. alba*, *P. adenopoda*, *P. tomentosa*, and *P. alba* × *P. glandulose*. The results show that the elite individual GM107 is more closely related to *P.* alba and *P. alba* × *P. glandulose* than to other species. Taking into account their separate geographical distributions, *P. alba* can be considered as the likely maternal ancestor of GM107, excluding the possibility of *P. alba* × *P. glandulose*.

### 3.7. Transcription Profiles of cp Genes

To investigate the transcriptome profiles of cp genes, we collected roots, stems, leaves, and petioles from tissue-cultured plantlets of *P. alba* var. *pyramidalis*, *P. tomentosa* GM15, the elite individual GM107, and *P. trichocarpa*, and performed RNA sequencing. The transcripts of 66 genes were detected among the 4 tissue types, among which 44 genes were categorized into a low-expression group and 22 into a high-expression group. The expression levels of almost all 66 genes were higher in leaves than in other tissues, except for *PtatpF* and *PtatpH*. Among these genes, *PtndhF*, *PtpsbA*, *PtpsbB*, *Ptrps14*, *PtatpF*, and *PtatpH* showed higher transcript abundance in all tissues than all other genes (Figure 8).

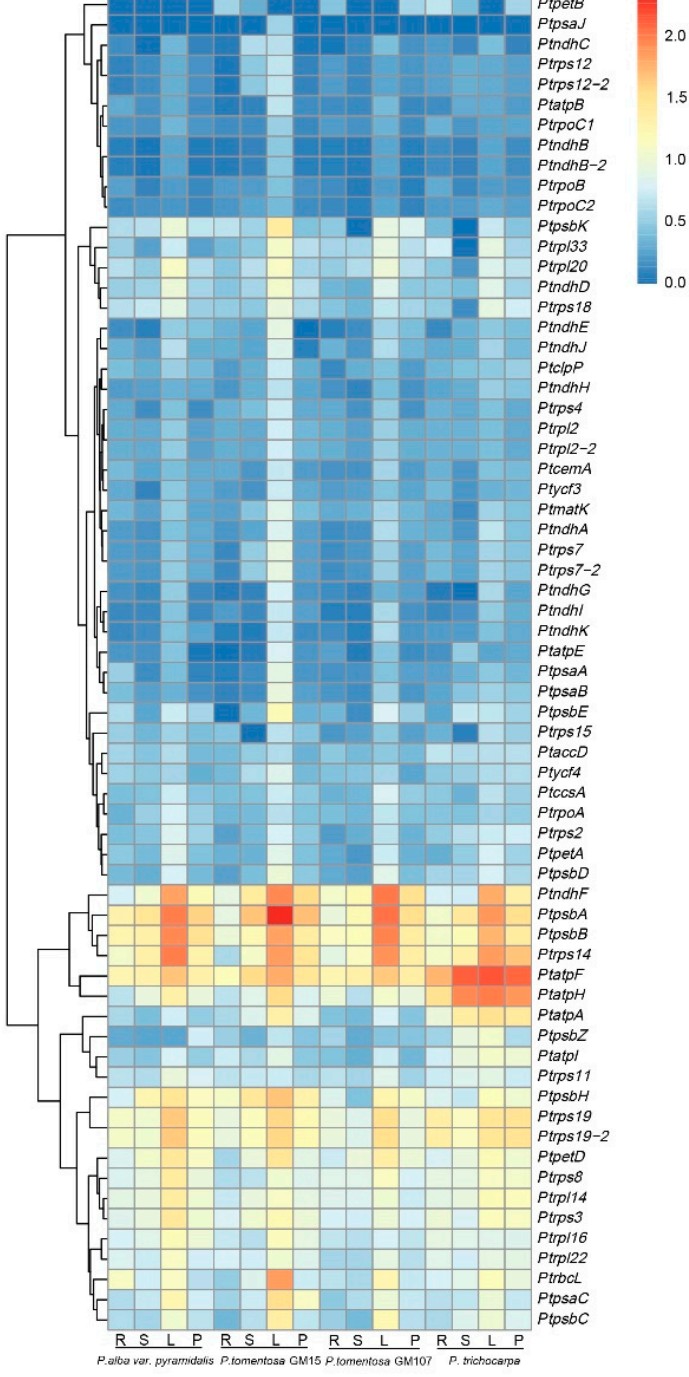

**Figure 8.** The transcription profiles of cp genes in roots, stems, leaves, and petioles from four poplars. R, S, L, and P represent roots, stems, leaves, and petioles, respectively.

## 4. Discussion

In this study, we presented the full chloroplast genome sequence of the elite individual GM107 and compared it to those of seven other white poplars. The results indicated that the quadruple structure, gene content, and organization of the cp genome were highly consistent, not only within poplar species [6,29,35], but also among other angiosperms [36,37]. The cp genome of GM107 was 156,493 bp long and encoded 127 genes, slightly different from other white poplars, which typically have a cp genome ranging from 155,355 to 158,591 bp and encode 131 genes, with the exceptions of *P. alba* (129) and *P. davidiana* (130).

Our research revealed minor variations in the positions of the IR and SC boundaries in different Populus species. These variations are a result of the expansion or contraction of the chloroplast genome, which is a common phenomenon among plants [37,38]. Compared with conserved boundary regions (except in *P. rotundifolia*) between the LSC and both IRs, the boundary region between the small single copy (SSC) and IRb was more variable among the eight poplar species. The *ycf1* gene, which is located on the SSC/IRa region, was relatively consistent among species, with a 1705 bp extension to IRa in 6 poplar species, and 2239 and 1002 bp extensions to IRa in *P. tremula* × *P. alba* and *P. rotundifolia*, respectively. By contrast, the *ycf1* gene, which is located on the SSC/IRb region, was more variable than that located on the SSC/IRa region (Figure 3). These findings suggest that changes in the IR region, mainly observed in white poplars, are due to variations in the SSC/IRb boundary.

We found 1034 SSRs in the chloroplast genomes of eight species of white poplar. The majority of these SSRs were A/T-rich and had a higher proportion of poly A/T than poly G/C. This aligns with the findings of previous studies [6,39,40]. Thus, SSRs located in the chloroplast genomes could be used as molecular markers for inferring phylogenetic relationships [41,42]. Large-scale overlapping of geographical distributions and easy crossing among white poplar species have led to abundant polymorphisms among section *Populus*. Although traditional phenotypic classification methods are intuitive and simple, they have some difficulty distinguishing hybrids or varieties with similar phenotypes and complex genetic backgrounds. SSRs in the cp genome are a powerful tool for the identification of white poplar species and their polymorphisms. DNA barcodes are effective molecular markers for species identification and genetic relationship analysis among species [43]. Generally, *nrITS*, *trnL-F*, *rbcL*, *matK*, and *trnH/psbA* are recommended for DNA barcoding in plants [44,45]. In this study, we identified 4 noncoding variable regions (*psbI-trnS*, *ndhC-trnV*, *petA-psbJ*, and *ycf1-ndhF*) showing >1% variation, making them potential candidates for use as molecular markers in phylogenetic analyses or DNA barcoding. Almost all species in section *Populus* are capable of interbreeding with at least one other species within the same section. This leads to a complex network of evolutionary change, making it challenging to distinguish between interspecific relationships [46] and identify hybrids and trace parental origins. Section *Populus* cp genomes show conserved quadruple structures and genes, have constant evolutionary rates [47,48], and, importantly, they follow maternal inheritance, which makes them suitable for inferring the phylogenetic status of the entire *Populus* section and tracing the parental origin of white poplar hybrids. In this study, we constructed a phylogenetic tree of the cp genomes of eight species in section *Populus* using ML methods. The results clearly indicated that interspecific relationships among section *Populus* species are highly stable. Notably, *P. alba* was found to be the maternal parent of the elite individual GM107. This finding differs from previously proposed origins for *P. tomentosa* [5,6,49]. There also appears to be variation within *P. tomentosa* with respect to its hybrid origin. In a previous study, *P. alba* was suggested to have played the role of the male parent species, with either *P. adenopoda* or *P. davidiana* as the female (maternal) parent [8]. However, in our recent study, *P. tomentosa* was suggested to have originated from hybridization between *P. adenopoda* (♀) and *P. alba* var. *pyramidalis* (♂) approximately 3.93 Mya [5]. Therefore, *P. tomentosa* may have multiple independent origins, suggesting a more complex evolutionary history.

## 5. Conclusions

In conclusion, the findings of this study provide valuable insights into the composition and organization of the cp genome of the elite individual GM107, a species that holds significant economic and ecological importance, and is widely found throughout China. We created a well-supported phylogenetic tree utilizing the existing chloroplast genomic information for section *Populus* and determined that the elite individual GM107 and *P. alba* were the most closely related species, followed by *P. alba* × *P. glandulosa*, which suggests that *P. alba* is the female parent of the elite individual GM107. These results will be helpful in elucidating the complex origin of Chinese white poplar and the phylogenetic relationships among poplar species in section *Populus*, and may contribute to future genetic improvements in the wood properties and carbon sink capacities of white poplar genetic resources.

**Author Contributions:** X.A. conceived the experiment. B.G., Y.L., T.C. and S.L. collected plant materials and conducted DNA and RNA extractions. R.-G.Z. assembled sequences and analyzed the data. K.-H.J. performed transcriptome analysis. X.A., B.G. and T.C. wrote and revised the manuscript. W.U.K. revised the manuscript. All authors have read and agreed to the published version of the manuscript.

**Funding:** This work was supported by Major Project of Agricultural Biological Breeding (2022ZD0401503), the National Natural Science Foundation of China (31870652, 31570661).

**Data Availability Statement:** The complete cp genome sequence of the elite individual GM107 cp sequence was submitted to the NCBI database with the GenBank accession number Pt OP894090.

**Conflicts of Interest:** The authors declare that they have no conflict of interest.

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
