# Peer review of "Comparative Analysis of the Characteristics, Phylogenetic Relationships of the Complete Chloroplast Genome, and Maternal Origin Track of White Poplar Interspecific Hybrid GM107"

_forests, doi:10.3390/f14030587_

Round 1

Reviewer 1 Report

The present study addresses the de novo sequencing and assembly of the whole chloroplast genome of white poplar hybrid GM 107 to understand the origin, evolutionary events in section Populus, and contribute to the molecular genetic improvement. The authors compare similarities and divergences of chloroplast genome traits and discussed the hypotheses regarding phylogenetic evolution, proposing molecular markers for inferring phylogenetic relationships. Overall, the manuscript is well structured and written. However, some points need more clarifications and can be improved specially in results and discussion as mentioned below:

My questions are mainly about some points in results that in my view are important and have not been discussed clearly.

“Results”

Pg. 7, line 211 and 212. What explains the significant variations in the JSB and JSA boundary sites between the species?

Some findings of the study highlight the divergent sequences of the non-coding regions. What explains these differences and what is the implication specially of “ycf1” for the Populus spp.?

Pg. 11, line 294. What the authors wanted to emphasize with codon usage frequencies "codon biased usage analysis" and Figure 6? This point should be better explained.

Line 313, In the phylogenetic analyses: there is some reason for cp genome of Populus trichocarpa not included? (Figure 7).  

By transcriptional analysis were identified highly expressed cp genome transcripts in different tissues. I guess to conform these profiles of expressed cp genes some of these could be validated by real-time qPCR and improve the findings of manuscript.

Author Response

1

Author Response

2

Round 2

Reviewer 2 Report

Thank you for your answer.

Author Response

Thank you for your comments. We have revised the language of the text and marked it in blue
